

# Prevalence of nasopharyngeal bacteria during naturally occurring bovine respiratory disease in commercial stocker cattle

Afroza Akter[1], Hannah Teddleton[1], Marc Caldwell[2], Gina Pighetti[1], Phillip R. Myer[1], Madison T. Henniger[1], Liesel Schneider[1] and Elizabeth Shepherd[1]

[1] Department of Animal Science, University of Tennessee-Knoxville, Knoxville, TN-Tennessee, United States
[2] Department of Large Animal Clinical Sciences/College of Veterinary Medicine, University of Tennessee-Knoxville, Knoxville, Tennessee, United States

Corresponding author
Elizabeth Shepherd,
eshephe4@utk.edu

## ABSTRACT

Bovine respiratory disease (BRD) is one of the most common economic and health challenges to the beef cattle industry. Prophylactic use of antimicrobial drugs can alter the microbial communities in the respiratory tract. Considering that the bovine upper respiratory tract microbiome has been associated with generalized health, understanding the microenvironment that influences this microbiome may provide insights into the pathogenesis of BRD. This study aimed to determine temporal variation in nasopharyngeal (NP) microbiome in naturally occurring BRD in newly received stocker calves. Mixed breed steers ($n = 40$) were purchased from an auction market and housed in a commercial stocker farm. Clinical signs were used to identify BRD affected animals, and calves were categorized based on the number of treatments (NumTrt) received (0, 1, 2). On days 0, 7, 14, and 21, NP samples were collected, and subsequent DNA were isolated and sequenced. After sequencing, 16S rRNA V4 gene was amplified and utilized for NP bacterial determination. The difference in relative abundance based on day and NumTrt was measured using repeated measures ANOVA (PROC GLIMMIX; SAS 9.4). Firmicutes, Proteobacteria, Actinobacteriota, Bacteroidota, and Verrucomicrobiota were the top phyla and *Mycoplasma*, *Histophilus*, *Geobacillus*, *Saccharococcus*, *Lactobacillus*, and *Pasteurella* were the top genera. In healthy calves, the relative abundance of *Mycoplasma* differed by day ($P = 0.01$), whereas on day 7, calves had five times greater abundance compared to day 0 (d 0: $0.06 \pm 0.05$; d 7: $0.30 \pm 0.05$). No differences were observed in the alpha diversity matrices based on day or NumTrt ($P > 0.05$). Results of this study suggest compositional variations in NP microbial populations occur during disease conditions.

## INTRODUCTION

Exposure to multiple risk factors, *i.e.*, recent weaning, transportation, commingling, castration, dietary change, and drastic weather/environmental variation, can affect the immune system adversely and increase risk for bovine respiratory disease (BRD) in beef cattle (*Taylor et al., 2010*). Approximately $54 million is spent annually in the US to treat BRD-infected animals, which can be a significant economic burden to producers, without factoring in associated production losses (*Johnson & Pendell, 2017*). Commingling of animals from multiple origins and unknown health statuses is a major source of viral infection causing inflammation in respiratory mucosa, reduced ciliary action and phagocytic activity, and colonization of opportunistic bacteria (*Edwards, 2010*; *Griffin, 2010*; *Timsit et al., 2016*). Although *Mannheimia haemolytica*, *Pasteurella multocida*, *Histophilus somni*, and *Mycoplasma bovis* are normal inhabitants of the bovine respiratory tract, under conditions where the host is stressed they may become opportunistically pathogenic and associated with BRD. Opportunistic pathogens such as *M. haemolytica*, *P. multocida*, *H. somni*, and *M. bovis* invade while animals are stressed, colonizing the respiratory tract to become the primary bacterial agents associated with BRD (*Klima et al., 2014*). Vaccination against BRD pathogens alongside prophylactic antimicrobial use are common management practices in high-risk cattle populations (*Anholt et al., 2017*; *Crosby et al., 2023*; *Credille et al., 2024*). However, widespread use of antimicrobials directed toward control of BRD is likely contributing to development of antimicrobial resistance (AMR) (*Klima et al., 2011*). Therefore, understanding nasopharyngeal microbial composition may contribute to better understanding of BRD development, diagnosis and respiratory health in stocker cattle.

Identifying other bacterial species associated with BRD pathogenesis is critical as uncharacterized pathogens may reduce efficacy of vaccination and antimicrobial treatment (*Johnston et al., 2017*). In previous studies, the most commonly abundant bacterial phyla isolated from feedlot cattle nasopharyngeal (NP) samples are *Proteobacteria*, *Firmicutes*, *Tenericutes*, and *Actinobacteria* (*Holman et al., 2015*, *2017*). However, bacterial load of the respiratory tract were compositionally different in BRD-infected feedlot cattle when compared to clinically healthy cattle (*Holman et al., 2015*). When assessing NP microbiota during a stressful stimulus, cattle compositions differed significantly prior to transport compared to two days post-transport (*Holman et al., 2017*), suggesting relative diversity of microbiota may be useful in understanding animal health status and later development of BRD. However, limited studies have focused on the temporal dynamics of NP bacterial composition in relation to treatment for naturally occurring BRD in commercial stocker cattle. Therefore, our objective was to investigate temporal dynamics of the NP microbiota in stocker cattle on arrival to a commercial stocker facility, and additionally to characterize differences in the NP microbiota of animals receiving multiple treatments for BRD, compared to calves that were not diagnosed or treated for BRD.

## MATERIALS AND METHODS

### Study population

All procedures were performed with the approval of a commercial stocker producer on privately managed calves and the University of Tennessee Institutional Animal Care and Use Committee (Protocol # 2878-1221). Calves were received from an order buyer and maintained on pasture at a commercial stocker operation in Crossville, TN. Calves were kept on 15-acre field with mixed forage containing native warm and cool season grasses. All calves were supplemented daily with a mix of crushed corn, corn gluten, and straw, as well as a pelleted calf starter (14% crude protein) (*Akter et al., 2022*). Calves were managed according to the farm's standard operating procedures. At conclusion of the study, because all calves were owned privately, calves remained at the stocker facility and maintained at the discretion of the producers.

Forty clinically healthy mixed-breed beef bull and steer calves (225 ± 25 kg) were selected for the study. Prior to arrival at the stocker facility, the order buyer administered broad-spectrum anthelmintics (12 mL Safe-Guard Suspension Drench; Merck Animal Health, Rahway, NJ, USA), antibiotics (7 mL Excede subcutaneously; Zoetis Inc., Parsippany-Troy Hills, NJ, USA), mineral injection (5 mL Multimin 90 sub-cutaneously; Multimin North America, Inc., Fort Collins, CO, USA), and vaccinations against BRD and clostridial pathogens (MpB Guard subcutaneously; American Animal Health Inc., Grand Prairie, TX, USA; Ultrabac 7 subcutaneously; Zoetis Inc., Parsippany-Troy Hills, NJ, USA), and Pyramid 5 + Presponse SQ subcutaneously (Boehringer Ingelheim Animal Health USA, Duluth, GA, USA). Ear notch samples were collected from all calves to test for persistent bovine viral diarrhea virus infections, with none testing positive. Intact bull calves were castrated ($n = 27$) and horns were removed as needed. No unexpected adverse events occurred during the study.

### Clinical observation of calves

Calves were monitored once daily for BRD clinical symptoms by the producer. A trained clinician scored each animal and assigned BRD diagnosis on sampling day 0, 7, 14 and 21 (relative to entry to the stocker facility). Clinical severity score (CSS) was recorded as CSS 0 (normal; no signs of clinical illness), CSS 1 (slightly ill; gaunt or nasal/ocular discharge), CSS 2 (moderately ill; any of the two following clinical signs: gaunt nasal/ocular discharge, coughing, labored breathing, and physically walking behind other animals in the group), CSS 3 (severely ill; purulent nasal/ocular discharge, severe labored breathing), and CSS 4 (moribund; non-responsive to human approach) (*Pillen et al., 2016*).

If diagnosed with BRD calves were given 40 mg/kg body weight of florfenicol (Nuflor; Merck Animal Health, Rahway, NJ, USA) subcutaneously and 2.2 mg/kg body weight of flunixin meglumine (Benamine; Merck Animal Health, Rahway, NJ, USA) intravenously. If BRD persisted on the subsequent sampling date, calves were administered subcutaneous tulathromycin (Draxxin; Zoetis Inc., Parsippany-Troy Hills, NJ, USA) with a dose of 2.5 mg/kg body weight. Calves were assigned a retroactive treatment status based on the

total number of antibiotic treatments received for BRD over the entire study period (NumTrt; 0×: never treated; 1×: treated one time; 2×: treated two times). Calves were assigned to category 0× (if they received no antibiotic treatments), 1× (if they received florfenicol and flunixin meglumine), and 2× (if they received tulathromycin on the consecutive sampling).

Lung consolidation was assessed by transthoracic ultrasound (TUS) for both left and right sides using a portable ultrasound. Isopropyl alcohol (70%) was used as a transducing agent and was applied between the 3$^{rd}$ and 12$^{th}$ intercostal space. A TUS score of 1 (normal; aerated lung with no consolidation), score of 2 (moderate consolidation, no more than two distinct areas of consolidation of <3 cm$^2$; diffuse comet-tail artifacts and lobular lesions) or score of 3 (severe consolidation; consolidation area ≥3 cm$^2$ and/or greater than two distinct areas of consolidation of <3 cm$^2$) was given to each calf (Akter et al., 2022). Calves with TUS score of 1 in both left and right lungs and no treatment for BRD on the sampling day were characterized as healthy. Calves with at least one lung scoring a TUS of 2 and no treatment were considered as subclinical, while those with a TUS of at least two in any of the lungs and received treatment for BRD were categorized as clinical.

Calves were categorized as BRD infected when they had CSS 1 with rectal temperature of >40 °C, or if they had a CSS >2, regardless of rectal temperature. TUS scoring was used as a measure of lung pathology and used in conjunction with CSS to diagnose BRD.

## Nasopharyngeal swab collection

Nasopharyngeal (NP) samples were collected using one swab per nare, on days 0, 7, 14, and 21. Nasopharyngeal (NP) swabs were collected using a 33-inch double guarded swab (Santa Cruz Animal Health, Dallas, TX, USA). Immediately prior to NP swab collection, external nares were wiped with an alcohol swab. The nasopharyngeal swab was inserted into the ventral meatus of the nose and rotated against the pharyngeal wall for 20–30 s. Swabs were removed and cotton tips were broken off into 5 mL conical tubes containing Brain Heart Infusion (BHI) and glycerol (1:1 of BHI and glycerol). Samples were transported on ice to the laboratory and processed on the sample day as sampling., Samples were vortexed, samples from left and right nares were pooled, and the liquid portion was stored at −80 °C for future analysis. Because swabbing was minimally invasive and causes little distress, no anesthetics were administered.

## DNA extraction and sequencing

For DNA extraction, a QIAamp® PowerFecal® DNA kit (Qiagen, Hilden, Germany) was utilized with modifications. Samples were thawed, centrifuged at 13,000× g for 5 min, and the supernatant was discarded. The subsequent pellet was resuspended in 180 µL lysis buffer (Hart et al., 2015), and incubated for 1 h (37 °C). After which, 25 µL proteinase K and buffer AL were added, followed by incubation for 30 min (56 °C). The supernatant was transferred to lysis tubes (Zymo Research Corp, Tustin, CA, USA), and cell lysis was performed utilizing a TissueLyser II system (Qiagen, Hilden, Germany) for 3 min at 21 Hz.

After which, 200 μL of 100% ethanol was added to the lysate, mixed by vortexing, and then transferred to an DNeasy spin column placed within a 2 mL collection tube (Qiagen, Hilden, Germany). The lysate was centrifuged at 6,000× *g* for 1 min (37 °C), and flow-through was discarded. After replacing the collection tube, 500 μL of buffer AW1 was added, and centrifuged for 1 min at 6,000× *g* for 1 min. After discarding flow-through and replacing the collection tube, 500 μL of buffer AW2 was added and centrifuged at 20,000× *g* for 3 min to dry the DNeasy membrane. The DNA was eluted by the addition of 200 μL of buffer AW directly to the DNeasy membrane, followed by centrifugation at 6,000× *g* for 1 min. DNA sample concentration and purity was measured using a spectrophotometer (DeNovix, Wilmington, DE, USA).

Extracted DNA was sent to the University of Tennessee Genomics Core laboratory facility for library preparation and amplicon sequencing. Library preparation and amplicon sequencing targeted the V4 hypervariable region of the 16S rRNA gene with modified adapters used for Illumina MiSeq sequencing (*Henniger et al., 2022*). Broad-spectrum forward (515Fb; *Parada, Needham & Fuhrman, 2016*) and reverse (806Rb; *Apprill et al., 2015*) primers were used and sequencing was conducted in a two-step polymerase chain reaction (PCR). Conditions of PCR were performed using the following thermal cycler profile: 10-min hold at 95 °C, 35 cycles of 95 °C for 30 s, 55 °C for 30 s, and 75 °C for 30 s, followed by 72 °C for 10 min. Purification of the PCR product was completed using 20 μL of AMPure XP beads (Beckman Coulter, Breca, CA, USA), with indexing of each sample utilizing a unique combination of forward and reverse Nextera XT v2 indexes (Illumina, San Diego, CA, USA). Library sequencing was completed using Illumina MiSeq (2 × 300 bp) and the MiSeq Reagent Kit v3 (600-cycle; Illumina, San Diego, CA, USA).

## Analysis of gene sequences

DADA2 v1.14 (*Callahan et al., 2016*) was used to process raw sequences of the 16S rRNA gene in R v1.4.1717 (*R Development Core Team, 2010*). Unless stated otherwise, DADA2 was used to process and analyze the gene sequences. Low quality sequences were identified, trimming was performed using Phred quality scores (trimmed if <30), and 250 and 200 base lengths were truncated in the forward and reverse reads, respectively. Error rate was determined using approximately 106 million forward and reverse reads. Forward and reverse reads were merged, and chimeric sequences were removed. Using a 97% similarity threshold value, all high-quality sequences were assigned to a specific operational taxonomic unit (OTU). Each OTU was analyzed against the SILVA 138.1 SSU taxonomic database (*Quast et al., 2013*) by naïve Bayesian RDP classifier method (*Wang et al., 2007*). Shannon diversity and Chao1 diversity index were measured to identify alpha diversity using Phyloseq v1.30.0 (*McMurdie & Holmes, 2013*) and vegan v2.5-7 (*Oksanen et al., 2013*) packages in R (*R Development Core Team, 2010*). Beta diversity analyses were performed using the Bray-Curtis dissimilarity in vegan. Principal coordinate analysis (PCoA) utilized the Bray-Curtis dissimilarity to determine beta diversity based on the TUS status at different sampling days.

## Statistical analyses

The GLIMMIX procedure in SAS 9.4 (SAS Institute Inc., Cary, NC, USA) was used to test differences in relative abundances occurring due to NumTrt and day. Temporal variation in alpha diversity matrices or relative abundances of the top phyla or genera within cattle never receiving treatment was measured by using the mixed model analysis of variance (ANOVA), where the fixed effect of sampling day and the random effect of individual calf ID with the random residual of the day were included. Additionally, variation in the alpha diversity on treatment days 14 and 21 were measured using the GLIMMIX procedure in SAS 9.4. The effect of Numtrt on alpha diversity matrices or relative abundances of top phyla or genera were analyzed using the mixed model ANOVA with the fixed effect of NumTrt, and the random effect of individual calf ID, with the random residual of the day. In a separate model, sampling day was considered the fixed effect, whereas individual calf ID was the random effect, with the random residual of the day included as an autoregressive repeated term. To measure differences in the bacterial community based on TUS status, permutational multivariate analysis of variance (PERMANOVA) was computed using the vegan package in R with 999 permutations (*R Development Core Team, 2010*). Differences in alpha diversity matrices or relative abundances of top phyla or genera between calves that were either healthy or received treatment on a given day were also evaluated. The fixed effect of treatment by day was evaluated by ANOVA. Since these analyses were subset by day, no random effects were required. For all the analyses, differences were considered significant when $P \leq 0.05$.

## RESULTS

Out of the study population of 40 calves, 16 calves were diagnosed at least once with clinical BRD, and 24 calves remained clinically healthy during the duration of the study period (cumulative incidence: 40%). On day 0, no calves were treated for BRD. On day 7 of the study, two calves were treated (TUS score 2; clinical). On day 14, 11 calves were treated for BRD (TUS score 2; clinical) and 12 calves received a TUS score of 2 (subclinical for BRD). On the last sampling day (d21) eight calves received antibiotics for BRD (TUS score 2; clinical), whereas 14 calves were given a TUS score of 2 (subclinical) but did not receive treatment. Of the 40 calves enrolled in the study, 24 calves never received treatment (0×), 11 calves received one antibiotic treatment (1×), and five calves received antibiotic treatment twice (2×). Calves receiving no treatment had a TUS score of 1.

Amplicon sequencing of the NP microbiome from the V4 region of the 16S rRNA gene provided 16,111,772 unique sequences after quality filtering and chimera removal. 1,180 OTUss were clustered from sequences to assign different taxa using the SILVA database. The top five phyla, regardless of day or treatment status, were *Firmicutes* (71.23%), *Proteobacteria* (18.72%), *Actinobacteriota* (6.6%), *Bacteroidota* (2.27%), and *Verrucomicrobiota* (0.25%), which accounted for 99.1% of total bacterial abundance, which also is reflected in both treatment status (Fig. 1A) and day (Fig. 1B) at the phyla level. It is important to note that calves that received antibiotic treatment for BRD still retained populations of *Mycoplasma* (40.54%), *Histophilus* (26.23%), *Geobacillus* (7.58%), *Bacillus* (6.67%), *Saccharococccus* (4.72%), *Lactobacillus* (3.13%), *Clostridium sensu stricto 7*

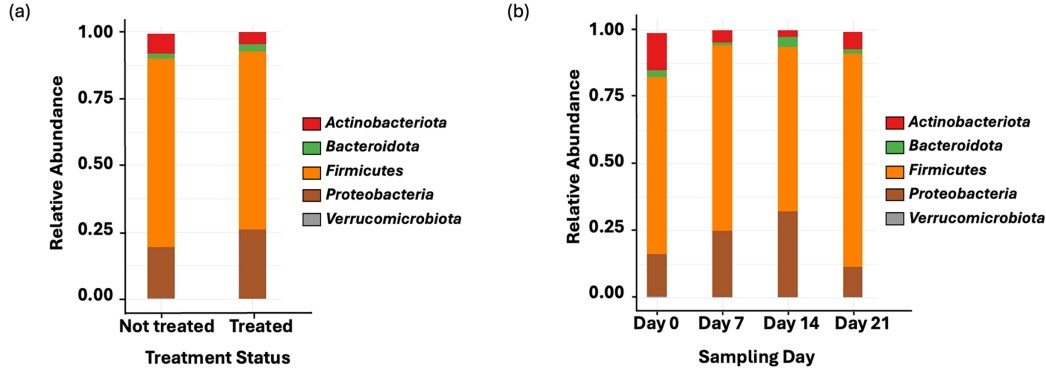

**Figure 1 Top bacterial phyla identified from nasopharyngeal samples.** (A) Relative abundance of bacteria phyla based on treatment status across all sampling days. Treated = 16 calves. Not treated = 24 calves. (B) Relative abundance of bacterial phyla on the nasopharyngeal sample cattle on sampling day 0, day 7, day 14, and day 21.               

(2.71%), *Filobacterium* (2.86), and *Mannheimia* (1%) (Fig. 2A). At arrival (day 0), all calves had reduced *Mycoplasma* (4.6%) compared to later sampling days (Fig. 2A). Apart from *Mycoplasma, Geobacillus* (20.39%), *Bacillus* (15.78%), *Saccharococcus* (12.05%), *Lactobacillus* (7.25%), *Moraxella* (4.62%), *Pasteurella* (4.42%), and *Acinetobacter* (2.91%) were present at day 0 (Fig. 2B). Interestingly, the abundance of *Mycoplasma* were 10 times higher (50%) at day 7, with *Histophilus* (2.84%), *Geobacillus* (9.91%), *Bacillus* (7.89%), *Saccharococcus* (5.92%), *Lactobacillus* (3.74%), *Moraxella* (2.96%), and *Pasteurella* (1.85%) as the abundant genera (Fig. 2B). On days 14 and 21, *Mycoplasma* (39.87%; 29.03%) and *Histophilus* (22.08%; 26.95%) were the most abundant genera (Fig. 2B). For *Actinobacteriota*, *Bacteroidota*, and *Verrucomicrobiota*, the greatest abundance occurred at Day 0, and no differences were observed among other days. A NumTrt by day interaction were observed for *Mycoplasma* ($P = 0.03$) (Fig. 3).

## Differences in alpha and beta diversity measures

Although bacterial composition differed based on TUS status, a low $R^2$ value indicates poor clustering (Fig. 4). Among the subset of clinically healthy calves, no differences in temporal variation from the Chao1 metric for the OTUs were observed ($P > 0.05$) (Table S1). Likewise, there was no temporal variation in the Shannon diversity index in the subset of healthy calves observed, except at day 14 ($P < 0.05$). Neither alpha diversity measures (Chao1 and Shannon diversity) differed based on treatment status at days 14 and 21 ($P > 0.05$). Based on beta diversity analyses, the composition of bacteria differed based on TUS staus ($P = 0.001$; $R^2 = 0.04$).

## Temporal changes of bacterial phyla and genera among healthy calves

Temporal differences in the relative abundance were observed in phyla Firmicutes ($P = 0.03$), Actinobacteriota ($P < 0.0001$), Bacteroidota ($P = 0.001$), and Verrucomicrobiota ($P = 0.001$) in healthy calves (Table S2). No differences were observed in the relative abundance for Proteobacteria ($P = 0.13$). Significant differences were observed in the relative abundance based on day for *Mycoplasma* ($P = 0.01$); however, no differences were

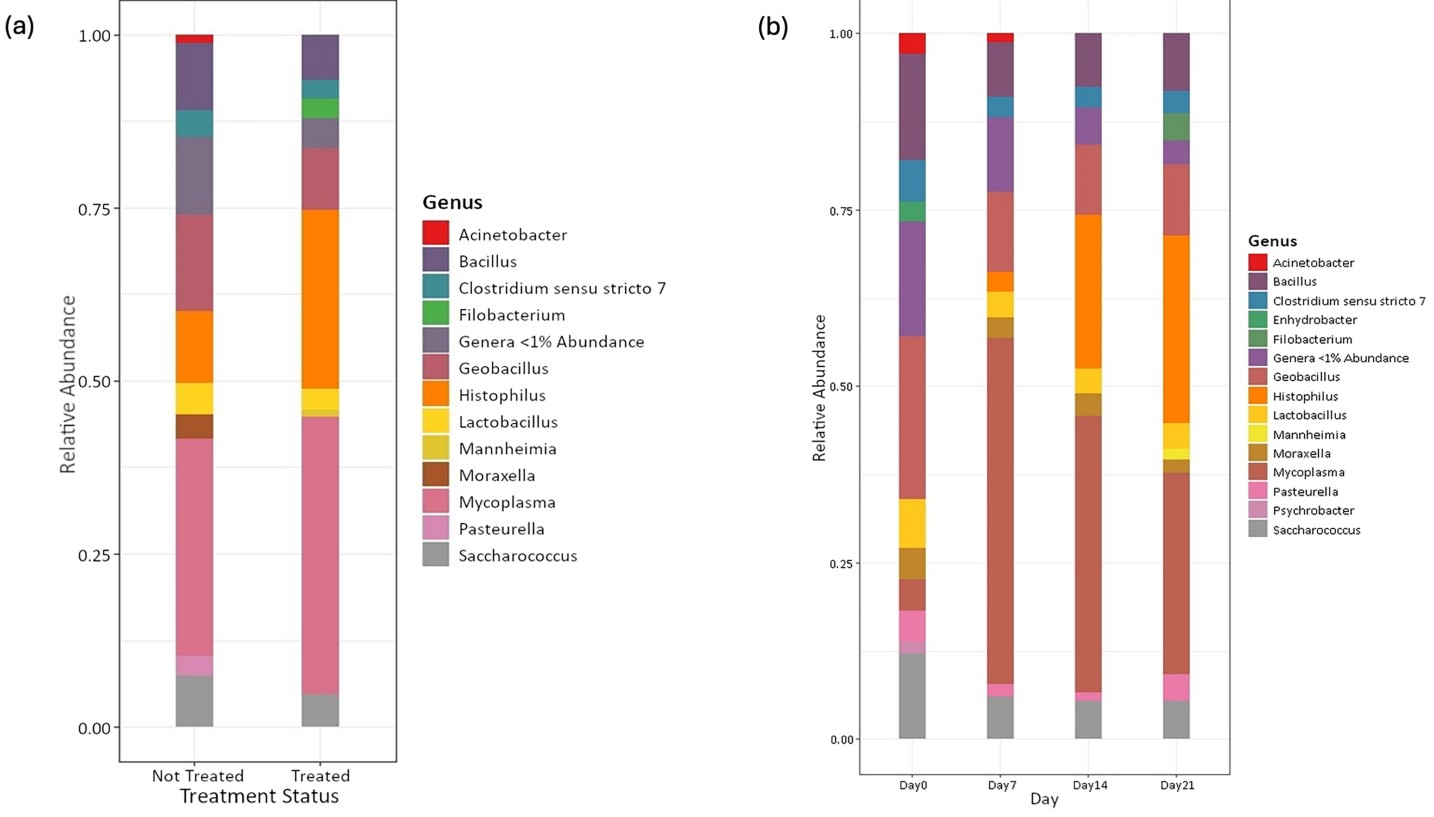

**Figure 2  Top bacterial phyla identified from nasopharyngeal samples.** Relative abundance of bacterial phyla of nasopharyngeal by (A) treatment status and (B) by sampling day 0, 7, 14, and 21.

observed for *Histophilus, Pasteurella*, *Bacillus*, and *Lactobacillus* ($P > 0.05$) in healthy calves (Table S3). In the clinically healthy calves, *Histophilus* was not identified within 1% abundance at arrival; however, on day 7, one calf had a population of *Histophilus* (Table S3). Based on the antibiotic treatment status, calves who received no treatment had a higher abundance of *Mycoplasma* (31.9%), *Histophilus* (10.56%), *Geobacillus* (12.15%), *Bacillus* (10.02%), *Saccharococccus* (7.27%), *Lactobacillus* (4.64%), *Clostridium sensu stricto 7* (3.85%), *Moraxella* (3.57%), and *Pasteurella* (3.04%) (Table S3).

## Relative abundance of bacterial phyla and genera based on treatment status

The effect of the number of antibiotic treatments given (NumTrt), day, and the two-way interaction on relative abundance for top bacterial phyla highlighted differences in Actinobacteriota ($P = 0.02$) based on NumTrt (Table S4). Calves with BRD receiving 1× antibiotic treatment had greater abundance of Actinobacteriota. Although calves receiving no treatment expressed numerically higher Actinobacteriota abundance, they were not statistically different from the calves receiving 1× or 2× antibiotic treatments (Table S4). Based on NumTrt, relative abundance of Firmicutes, Proteobacteria, Bacteroidota, and Verrucomicrobiota did not differ ($P > 0.05$) (Table S4). A day effect occurred with relative abundance of Proteobacteria ($P = 0.03$), Actinobacteriota ($P < 0.0001$), Bacteroidota

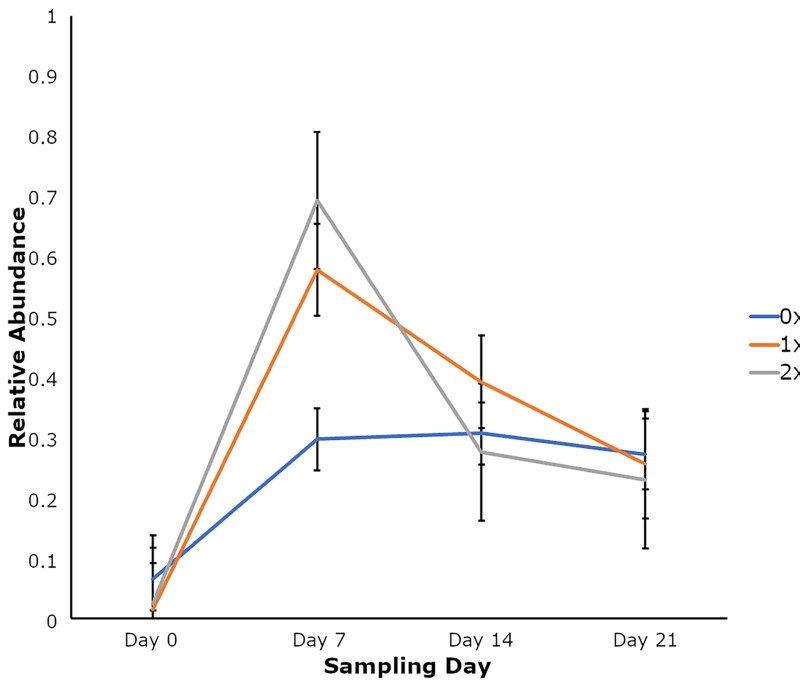

**Figure 3 Relative abundance of Mycoplasma based on the number of antimicrobial treatments.**
Relative abundance of Mycoplasma based on the number of antimicrobial treatments (NumTrt1) and
day 2 (NumTrt′ day; $P$ = 0.03) of newly received stocker cattle. 1NumTrt was defined based on the
number of antibiotic treatments they received (0×: never treated; 1×: treated once; 2×: treated 2 times).
2Day 0, Day 7, Day 14, and Day 21 denote the day relative to calves' arrival to the stocker farm.

($P$ = 0.002), and Verrucomicrobiota ($P$ = 0.0001) (Table S4). At day 7, clinically healthy
calves had significantly reduced *Mycoplasma* than 1× and 2× antibiotic treated calves;
however, no differences were observed among NumTrt status for other days (Fig. 4). Effect
of NumTrt were observed for *Pasteurella* ($P$ = 0.04) and *Bacillus* ($P$ = 0.03) (Table S5).
Based on NumTrt, *Pasteurella* were significantly higher in the clinically healthy calves,
whereas the 2× antibiotic treated group had significantly lower *Bacillus* (Table S5).
Significant effect of day was observed for *Histophilus* ($P$ < 0.0001), *Lactobacillus*
($P$ = 0.001), and *Bacillus* ($P$ = 0.007) (Table S5).

## DISCUSSION

Previous studies have demonstrated temporal shifts in NP microbiota in transitioning
cattle arriving at feedlots, with increases in diversity and richness, where NP microbes
stabilized and became more homogenous after 60 days on the feedlot (*Holman et al., 2015,
2017*). Stability and shifts of respiratory microbial populations during transition have been
linked to disease risk and development in stressed cattle (*Zeineldin et al., 2017*; *Timsit et al.,
2018*). Within the current study, both healthy cattle and cattle treated with antibiotics
showed differences in phyla and genera over time; however, healthy cattle showed fewer
changes in genera (*Mycoplasma*) across the sampling period, suggesting greater stability in
NP microbial communities compared to treated cattle. Although clinically healthy calves
did not receive any antimicrobials during the study period, administration of

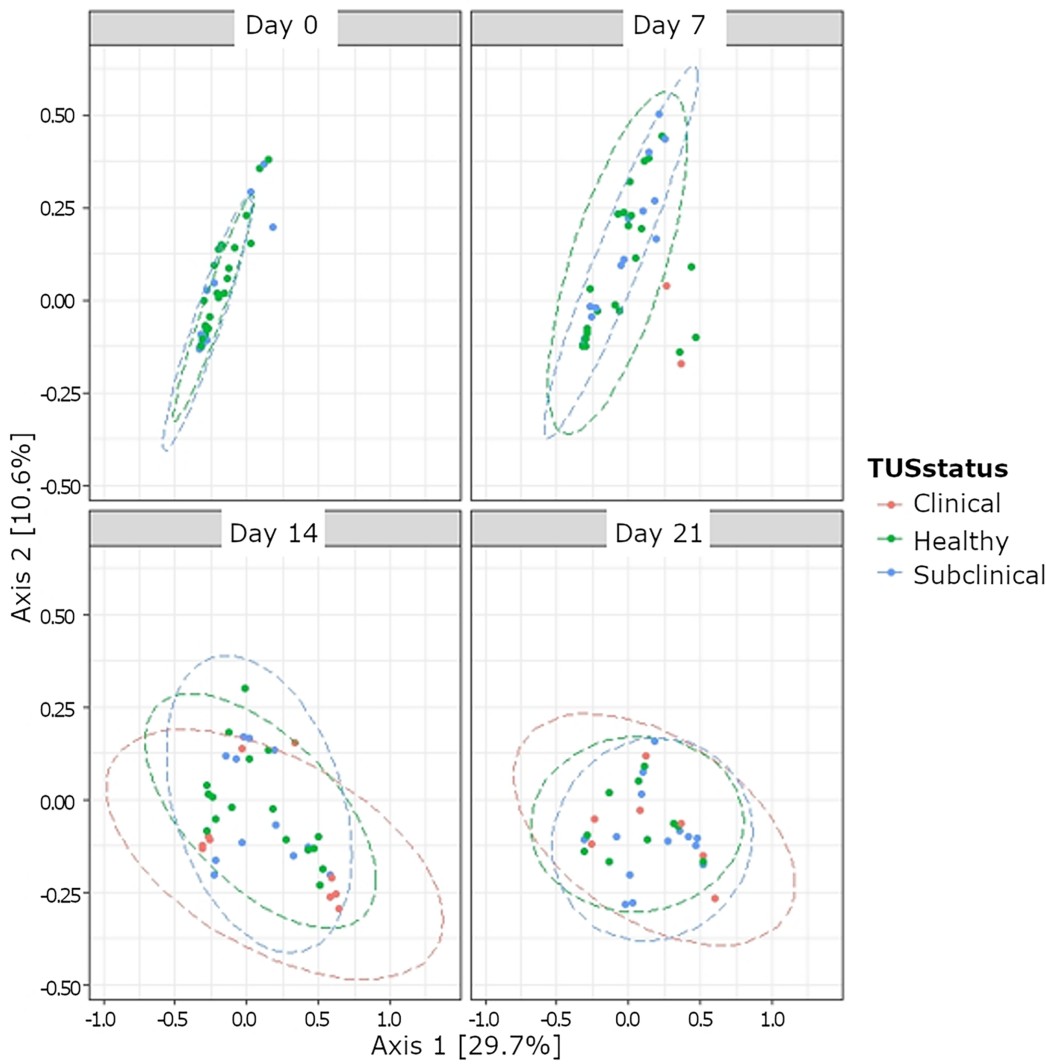

**Figure 4 Beta diversity of nasopharyngeal bacteria based thoracic ultrasonography (TUS status) and day based on BRD treatment status.** TUS scoring on sampling day 0, 7, 14, and 21. Healthy calves received a TUS score = 1. Subclinical BRD TUS score = 2, but calves did not receive treatment based on CSS. Clinical BRD TUS score = 2 and calves received treatment (CSS = 1 and body temperature >40 °C, or CSS > 2).

broad-spectrum antimicrobials prior to sampling could have influenced microbial colonization to the upper respiratory tract (*Holman, Yang & Alexander, 2019*), decreasing abundance of microbes. Furthermore, Ceftiofur (given prior to sampling) is ineffective in treatment of *Mycoplasma* (*Cai et al., 2019*), which may explain why Mycoplasma did not differ across sampling period and stabilized as a dominate genera.

Identifying common genera across healthy cattle varies, because microbial populations can differ between upper and lower respiratory tracts (*Zeineldin et al., 2017*). Additionally, environmental changes, such as transitioning to feedlots (*Timsit et al., 2018*) can influence microbial communities. Within the current study, the predominant genera included

*Mycoplasma, Histophilus, Geobacillus, Bacillus, Saccharococcus, Lactobacillus, Moraxella,* and *Pasteurella*, which differs from previously reported stocker cattle that identified an abundance of *Bacillus, Staphylococcus, Moraxella, Pasteurella* and *Mannheimia* (*Holman, Timsit & Alexander, 2015*). *Lactobacillaceae* in the narsopharynx was associated with clinically healthy cattle (*Holman et al., 2015*); however, *Lactobacillacea* has been identified in cattle recovered from BRD (*Centeno-Martinez et al., 2023*). In the current study, *Mycoplasma* was associated with healthy cattle at day 7, whereas no changes in *Lactobacillaecea* occurred over time or with treatment. Differences in dominant genera may be due to short sampling period, or prophylactic treatment prior to enrollment in the current study, rather than transportation or changing environment.

Temporal changes in *Mycoplasma* were observed in healthy cattle, whereas abundance of *Mycoplasma* did not change in cattle treated for BRD. In feedlot cattle with stabilized NP microbial communities, *Mycoplsma* accounted for 14.9% of isolated bacteria from NP swabs (*Holman, Timsit & Alexander, 2015*). While *Mycoplasma dispar* and *Mycoplasma bovis* accounted for 53% of NP bacterial genera in clinically healthy cattle, we could not resolve *Mycoplasma* species in our data and therefore, it is difficult to attribute pathogenesis of BRD to specific *Mycoplasma* species. Although *Mycoplasma* has been associated with BRD in prior studies, *Mycoplasma* may be present in clinically healthy cattle, but becomes opportunistic colonizing with BRD progression (*Allen et al., 1991*). Prophylactic treatment with antibiotics prior to enrollment in the study may have impacted Mycoplasma abundance. Additionally, microbial communities identified in the nares may not be reflective of microbial communities present in the lungs during BRD infection. To control for factors such as breed, age, environment, and administration of antimicrobials, will improve understanding of host-microbiome interactions and disease pathogenesis.

Interestingly, upon arrival *Mannheimia* was not detectable in NP samples and no calves were treated for BRD on day 0. Typically, *Mannheimia* is a commensal bacterium of the upper respiratory tract (*Rice et al., 2007*) but was not detectable until day 21 in treated calves. Later detection of *Mannheimia* may suggest opportunist infection due to compromised host defenses after multiple treatments, or when high levels of mycoplasma are present (*Rice et al., 2007*), rather than stress of transport or mixing at the stocker facility.

In terms of diversity, previous research in feedlot cattle indicated BRD cattle had lower bacterial diversity in NP samples after feedlot arrival (*Holman et al., 2015*), which may indicate progression of pathogenic bacterial colonization after a stressful stimulus or increased risk factors. However, other studies suggest variability in prevalence of bacteria over time, with different metaphylaxis options (*Abi Younes et al., 2024*). Of the identified microbes associated with BRD treatment, *Pasteurella* and *Bacillus* changed over time and with treatment, which is in alignment with other studies that have identified *Pasteurella sp* associated with BRD pathogenesis (*Allen et al., 1991*; *Angen et al., 2009*; *Holman et al., 2015*), indicating these bacterial genera may be associated with BRD status. However, overall bacterial diversity did not differ temporally within the current study, which may

indicate instability in bacterial colonization in a 20-day period, or a result of a small sampling size. Furthermore, microbial communities may have been impacted by administration of antibiotic treatment prior to the start of the sampling period, resulting in unstable microbial communities. *Mycoplasma* has documented resistance to florfenicol (*Bokma et al., 2020*), and thus may have attributed to increased *Mycoplasma* following treatment in BRD cattle. Conversely, *Histophilus* were most abundant after *Mycoplasma* in BRD-treated cattle, which persisted following 2× treatment, despite reported efficacy in treatment of BRD (*Thiry et al., 2014*; *Bokma et al., 2020*). Thus, it is unclear the effects of treatment on NP microbial populations. Monitoring BRD incidence and bacterial abundance may provide greater insight into BRD progression; however, as a commercial farm and the lack of consent for extended monitoring, it was not feasible to continue observing the calves beyond 4 weeks. Understanding changes in NP microbial populations may also provide insight into impacts of antibiotic drug use on NP health.

## CONCLUSIONS

At phylum level, Firmicutes, Proteobacteria, Actinobacteriota, Bacteroidota, and Verrucomicrobiota were the most dominant NP bacteria in stocker calves in relation to the prophylaxis provided. *Mycoplasma* were the most abundant genus regardless of day and treatment status. Variation in bacterial diversity was observed between treated and non-treated cattle; however, diversity did not differ between treated groups. Individual variation in NP microbiome abundances was also observed. In consideration of the limitations of the study, *i.e.*, difference in genetics, age of the calf, limited number of calves in each treatment status, use of different antibiotics, and shorter follow-up periods, further respective studies are critical to understand temporal patterns of NP microbiome in high-risk stocker cattle.

## ACKNOWLEDGEMENTS

The authors would like to express thanks to Dennis Toc Mo, and the animal science graduate students for helping in the data collection.

### Funding

This project was supported by the University of Tennessee Institute of Agriculture (UTIA) through the state of Tennessee, the Department of Animal Science, and the United States Department of Agriculture National Institute of Food and Agriculture Multistate project NC1192. Funding was managed by UTCVM Center of Excellence in Livestock Diseases and the Human Health competitive grant. The funders had no role in study design, data collection and analysis, decision to publish, or preparation of the manuscript.

### Grant Disclosures

The following grant information was disclosed by the authors:
University of Tennessee Institute of Agriculture (UTIA) through the State of Tennessee.

Department of Animal Science, and the United States Department of Agriculture National Institute of Food and Agriculture Multistate: NC1192.

UTCVM Center of Excellence in Livestock Diseases and the Human Health Competitive Grant.

## Competing Interests

The authors declare that they have no competing interests.

## Author Contributions

- Afroza Akter conceived and designed the experiments, performed the experiments, analyzed the data, prepared figures and/or tables, authored or reviewed drafts of the article, and approved the final draft.
- Hannah Teddleton analyzed the data, prepared figures and/or tables, authored or reviewed drafts of the article, and approved the final draft.
- Marc Caldwell conceived and designed the experiments, authored or reviewed drafts of the article, and approved the final draft.
- Gina Pighetti conceived and designed the experiments, authored or reviewed drafts of the article, and approved the final draft.
- Phillip R. Myer conceived and designed the experiments, analyzed the data, prepared figures and/or tables, authored or reviewed drafts of the article, and approved the final draft.
- Madison T. Henniger analyzed the data, prepared figures and/or tables, and approved the final draft.
- Liesel Schneider conceived and designed the experiments, analyzed the data, authored or reviewed drafts of the article, and approved the final draft.
- Elizabeth Shepherd conceived and designed the experiments, analyzed the data, prepared figures and/or tables, authored or reviewed drafts of the article, and approved the final draft.

## Animal Ethics

The following information was supplied relating to ethical approvals (*i.e.*, approving body and any reference numbers):

All procedures were performed with the approval of a commercial stocker producer on privately managed calves and the University of Tennessee Institutional Animal Care and Use Committee (Protocol # 2878-1221).

## Data Availability

The sequencing data are available at NCBI: PRJNA1168492.

## Supplemental Information

Supplemental information for this article can be found online at http://dx.doi.org/10.7717/peerj.18858#supplemental-information.

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
