# Peer review of "Prevalence of nasopharyngeal bacteria during naturally occurring bovine respiratory disease in commercial stocker cattle"

_PeerJ, doi:10.7717/peerj.18858_

## Round 0.1 · original submission · Major Revisions

Dear authors, I ask you to pay attention to the reviewers' fundamental comments and make significant changes and additions to this manuscript. I hope that this will allow the reviewers to approve the publication of this article.

Reviewer 1 ·

Basic reporting

Prevalence of nasopharyngeal bacteria during naturally occurring bovine respiratory disease in
commercial stocker cattle (#107203)
I have provided specific comments below.
In general, the manuscript needs extensive editing. There are many instances where sentences are difficult to comprehend unless read multiple times – some examples have been flagged.
The study objective is not clear.
The M&M need more detail and it is unclear why the calves were analyzed by TUS status because there is no mention of this analysis in the Discussion. A major issue is whether the authors are reporting OTUs or ASVs - they appear to be used interchangeably, which is major error. There is also confusion with the study days and when the study truly started.
The Results need to be condensed. There is far too many Figures and Tables repeating what has been oultined in the text of the Results section.
Discussion – I have difficulty following the discussion. The only interesting findings to me were the high abundance of mycoplasmas and the lack of Mannheimia spp. on-arrival. I don’t think these findings were discussed within the context of other studies, or even speculated on why they anomolous results appeared. Were the just statistical aberrations due to a low number of animals? Was there a bias in how the data were generated/analyzed?

Abstract
- Line 28 – did you use multiple 16S rRNA genes, or the single 16S rRNA gene?
Introduction
- Line 44 – change to “$54 million is spent..”, should be singular not plural
- Line 46 – change to Commingling, comingling is the incorrect spelling
- Line 74 to 76 – I find the study objective a bit confusing. You note that Holman et al looked at the microbiota of NP post-transportation and that the diversity could inform animal health and BRD development – which might be overreaching on what they could actually predict. Then you note there are few studies on how the microbiota changes post-treatment. However, your study objective seems to be start out as a repeat of Holman et al with post-transportation NP analysis, but also differences in NP post-treatment. Do you have two separate objectives? Of is there one objective dedicated to pre- and post-treatment changes. You do not specify what is meant by “characterize differences in NO microbiota in animals treated for BRD”. What differences are you referring to – between pre and post-treatment, between treated and controls? And change “microbiota in animals” to “microbiota of animals”.

Materials and Methods
Study Population – be consistent in use of capitalization for subheadings
- Line 83 – add in the distance from the auction to the stocker operation – was it across the road or 300 miles? And can you describe the transportation – large cattle liners holding 80+ calves or smaller trailers? This is a ‘transportation study’ and hence more information on transportation and handling than is typically required for a BRD study should be included.
- Do you have any animal information on on the source of the calves. Could they have come from 1 or 2 cow-calf operations or do they represent a diverse group of operations?
- Line 91 – 92 – rewrite this line without the use of ‘where’, which makes it difficult to understand. I would even consider deleting this sentence because it is somewhat irrelevant that you examined 50 head. Easier to understand if just state “Forty clinically healthy ….. without evidence of injury or illness and having a body temperature of ≤ 40°C were selected for the study.”
- Line 93 – in most large commercial feedlots in North America, processing is done on arrival to the feedlot. You need to clarify that the processing was done at the auction prior to shipping. It is noted that this was Day 1 of the study. Perhaps it should be Day – 1 since the study has not yet started – you did not select the calves for the study until they arrived at the feedlot.

- Line 101 – BVD testing result should be placed in the Results section.
Clinical observation of calves
- Line 105 – specify if monitored once or twice daily.
- Line 105 – specify what were the sampling days.
- Line 110 – this is confusing. Four levels of health are described: normal, slightly ill, moderately ill, and severely ill. Then a fifth description is added – are these calves also categorized as slightly ill? Do all the calves, regardless of category, need to be febrile (> 40C). This section should be clarified. It is also confusing why the moderately ill definition uses is a choice of 3 clinical signs.
- Line 113 – the use of an acronym for the first time needs to be written out in full – TUS scoring?
- Line 115 – Could calves receive florfenicol if the day before they received Excede?
- Line 127 – how is this information different than what you stated on line 119-120.

Nasopharyngeal swab collection
- Line 132. Earlier it was stated that Day 1 was the day they were processed at the auction. If this is correct, then when are days 0, 7, 14 and 21?
- Line 135 – do you mean one swab from each nares or both swabs were inserted in each nares? Is there a reason you sampled both and then pooled?
- Line 139 – were the samples processed the same day as the collection or were they stored in a fridge until the next day?
DNA extraction
- I think you could logically group DNA extraction and the PCR and sequencing under one subheading.
- Line 166 – the line “The first step in the PCR… .” is incorrect. BSA would not be used to target primers.
- Line 184 – OTUs have been largely replaced with Amplicon Sequence Variants (ASVs), which provides better granularity to the data and allows comparisons between studies. I think this is a real weakness to this study, but I will leave other more knowledgeable reviewers to comment further.
Results
- Lines 216-220. The M&M states that the calves were also categorized by TUS status (healthy, sublcinical and clinical). How does the TUS system fit into your Results?
- Line 223 – earlier you mentioned OTUs and now you mention ASVs, these are two very different analysis and this needs to be clarified.
- Line 232 – it is not necessary and perhaps confusing to state “regardless of treatment status” because at Day 0 all the animals have the same status because none have been treated.
- Line 245 – you need to include a summary of TUS status at the beginning of the Results and clarification that the samples were analyzed by treatment status and also by TUS status.
- Line 294 to 297 – this sentence is confusing and should be separated into at least two sentences. Correct the spelling of commingling (line 296).
- Line 302 to 304 – as above this ‘run on’ sentence is difficult to understand without having to reread.
- Paragraph 302 to 311 – you have identified differences between your study and Holman et al (2015) with respect to predominant genera, but you give no explanation or speculation as to why they are different.
- Line 313 to 315 – I do not think that abundance alone is an indicator of whether a pathogen is associated with the BRD complex. Perhaps the antimicrobial therapy is why mycoplasmas were less abundant in the nares of treated cattle. And, what is happening in the nares may not necessarily be representative of what is occurring in the lungs, particularly with a complex disease such as BRD.
- Line 335 and 336 – why does instability lead to less bacterial diversity?
- Line 346 to 347 – do you mean reduced MH at 60 days versus at time of arrival to the feedlot? I don’t understand the comparison of reduced MH at 60 d to treated cattle? It is a mix of time and treatments.
- Line 347 – Holman et al (2018) found MH was relatively abundant at the the time of arrival, which is expected because MH is responsible for fibrinous pneumonia, which typically occurs within 10 -14 days of arrival, depending on metaphylaxis.
- The Discussion makes no mention of the association between the UTS status and microbiota.

Figures – all the titles are confusing because the wording “newly received stocker cattle” implies the data is actually generated from cattle at the time of arrival to the feedlot. Deleted newly received.
Figures1 and 2 do not convey much useful information or at least no more than what is written in the text. They could be combined with bar charts showing treated and nontreated beside each other at each time point.
Figures 3 and 4 could also be combined.
Figure 5 – again, not sure this adds much. Either decrease the text or delete the figure. It is redundant to show the figure and then describe it in detail in the text. And, if keeping the graph then you should denote any significant differences between the treatments by sampling day (ie. Is 1X significantly different than 2x at day 14?)
Figure 6 – the TUS status has already been described in the text. The legend needs to orientate the reader to what is contained in the graphic.
Tables – all five tables could be supplementary materials because the key results are contained in the Results section as text.

Experimental design

The research question is ill-defined and the M&M were not sufficiently detailed. I am not sure if all the calves came from a limited number of producers or were sufficiently mixed at the auction market. There is confusion as to the study days, type of analysis that was performed (OTUs of ASVs), and why the cattle were categorized by thoracic lung scoring.

Validity of the findings

How the data were generated needs to be clarified - OTUs or ASVs?
Overall, the lack of animals (samples) brings into question the validity of the data. Specifically, there were more healthy cattle (n = 24) than calves with a putative BRD diagnosis (n = 16). Furthermore, these 16 calves were further divided into a 1x (n = 11) and 2x (n = 5) treatment group.
The study should have been designed to recruit a specific number of BRD calves, with this number being determined by a power calculation or based on the previous studies referenced in the manuscript.

Additional comments

I think some caution is warranted when comparing the microbiota of stocker calves to high-risk calves entering into large commercial feedlots as was the case in the Holman studies. In the latter studies, the calves are reared under intensive conditions versus housed on 15 acre paddocks. And the mixing of the cattle at the auctions was probably more intense as was the stress of transport, weather, processing etc.

Reviewer 2 ·

Basic reporting

Line 58: This reference is outdated. More appropriate references would be Crosby et al, 2023 and Credille et al, 2024. These references show that anitmicrobial use is a factor in resistance but no the only driver.

Experimental design

Lines 85-86: How much supplemental feed was provided to the cattle daily and how was it fed? Do the authors have any data on the nutrient make up of the commodity blend feed?

Line 92: How were the 40 calves chosen from the total population of 50?

Line 95: Ceftiofur was administered to each calf 24 hrs prior to arrival. How do the authors think this might impact the results of the study? By definition, every calf received an antimicrobial treatment and, thus, no calf remained truly untreated? Could this have impacted the prevalence of Mycoplasma spp seen at the later sampling time points?

Line 95: how long following arrival were animals eligible for treatment given that they received ceftiofur 24 hrs prior to the study beginning?

Lines 101 and 102: How were castration and dehorning performed?

Line 104: Could the authors please cite a reference for this scoring system? It seems similar to the DART and CIS systems commonly used but as laid out here, it's a bit confusing. Please further clarify. I am also not clear as to how TUS was used in this study. Please elaborate further in the methods as to how it was implemented.

Line 105: The producer was scoring calves daily, while the research team was evaluating animals weekly....If animals were identified as having BRD by the producer, were they treated by him or were animals only treated if they were identified as having BRD on the weekly sampling dates? If this is the case, how could this have affected the results of the study?

Lines 105-106: Does the weekly retreatment mean that the post-treatment interval was by default 7 days?

Could the authors provide more detail on what proportion of animals were diagnosed as ill on what sampling days?

Validity of the findings

The authors will need to further clarify the design of this study. Quite a bit of information is unclear and, as it is currently written, it would be difficult to replicate this study.

Additional comments

Lined 340-343: The authors that Mycoplasma and H. somni are susceptible to flunixin meglumine....This is a bit perplexing. Based on the references provided do they mean florfenicol-flunixin? Please clarify

Lines 343-346: In populations such as this, the prevalence of BRD drops to almost 0 within 30 days of arrival. In other populations (low-risk cattle in feedlots), this might be different but I'm not sure that following animals further would have made much of a difference in a high-risk population of stocker calves.

How do the authors believe that the findings of this study could be used to impact the development of more effective treatment protocols in high-risk beef stocker calves?

Reviewer 3 ·

Basic reporting

It would be helpful to readers if the authors add a brief summary of the methods used for transthoracic ultrasound, so they don't need to retrieve Akter et al., 2022, to understand the results, since bacterial composition was related to TUS status.


Line 105: Because symptoms are reported by the patient, while signs are observed by the clinician, "symptoms" should be changed to "signs".

Line 110: What does "slightly ill" mean? Please modify to be more precise. How is it related to the signs described in the preceding sentences?

Line 113: "TUS" needs to be defined as this is the first time the acronym has been used.

Line 292: change to "...differences IN phyla and genera..."

Line 331 and 333: "Pasteurella" is misspelled

Line 341: change to "...susceptible to FLORFENICOL WITH flunixin meglumine..."

Line 395: the year of the Bokma et al reference needs to be corrected


Figures 1 and 3: As presented, it is no clear which sampling day is represented. Is this the difference between treated and not treated cattle at arrival? This should be clarified in the legend so the reader can easily understand the figures without going back to the text. Also, in the legend, it may be more correct to say "regardless of prior treatment" than "irrespective to prior treatment"

Figure 6: Where the legend says "A TUS score of at least 2 with and received treatment..." is a bit unclear, can you change the wording to make it more clear?


References: the authors should consider including a brief discussion and comparison with their findings of some relevant recently published papers that they have not referenced:


Abi Younes JN et al., Front Vet Sci. 2024 Jul 23;11:1416436. doi: 10.3389/fvets.2024.1416436. eCollection 2024. PMID: 39109351

Centeno-Martinez RE, Klopp RN, Koziol J, Boerman JP, Johnson TA. Front Vet Sci. 2023 Nov 16;10:1297158. doi: 10.3389/fvets.2023.1297158. eCollection 2023. PMID: 38033643

Howe S, Kegley B, Powell J, Chen S, Zhao J. Front Cell Infect Microbiol. 2023 Sep 8;13:1223090. doi: 10.3389/fcimb.2023.1223090. eCollection 2023. PMID: 37743862

McAtee TB, Pinnell LJ, Powledge SA, Wolfe CA, Morley PS, Richeson JT. Front Microbiol. 2023 Jun 13;14:1203498. doi: 10.3389/fmicb.2023.1203498. eCollection 2023. PMID: 37383638

McMullen C, Alexander TW, Orsel K, Timsit E. Vet Microbiol. 2020 Sep;248:108826. doi: 10.1016/j.vetmic.2020.108826. Epub 2020 Aug 17. PMID: 32891954

Experimental design

No comment.

Validity of the findings

This report describes the nasopharyngeal microbiome in recently received stocker cattle, and the relationship between BRD treatment and changes in the nasopharyngeal microbiome over 21 days after receipt of cattle. Descriptions of the nasopharyngeal microbiome in newly received cattle have been reported by others. However, due to the fact that the cattle in this study were managed on grass, and thus in a more extensive setting than a traditional feedlot, the findings merit reporting. The relationship between the nasopharyngeal microbiome and transthoracic ultrasound findings is also of interest. The relatively small number of cattle sampled, a limitation acknowledged by the authors, may limit external validity of the work.

This reviewer was not able to access the raw data using the link provided. Can the authors confirm that the raw data posted with NCBI are actually accessible?

Additional comments

No comment.

---

## Round 0.2 · accepted · Accept

I congratulate you on the acceptance of this article for publication. I hope you will continue your research in this direction.

Reviewer 1 ·

Basic reporting

The authors have satisfactorily addressed all my concerns. I would vote to accept the revised manuscript.

Experimental design

As above

Validity of the findings

As above

Additional comments

Nothing more to add.